# RoomGen: Text-Driven Agentic 3D In-door Scene Synthesis and Editing

## Abstract

Interactive 3D indoor scene generation is a crucial task with applications in embodied AI, virtual reality, and physics-based simulation. To enable the generated scenes that can be directly imported into off-the-shelf 3D engines, most prior work follows a retrieve-then-place pipeline. These systems typically combine large language models with traditional procedural content generation pipelines. While effective for one-shot generation of complete scenes, they lack incremental editability: inserting a new object often triggers global re-optimization, and localized re-layout is not natively supported. Moreover, most methods produce a semantic scene graph via an LLM, ignoring visual cues that naturally encode spatial relations. In this paper, we present an agent-based approach to scene layout generation that places objects sequentially. Conditioned on user instructions, we first retrieve relevant 3D assets, then iteratively select an object, predict its position and orientation, and place it in the scene. Each decision is conditioned on the current scene state, enabling flexible placement and incremental editing, including object insertion and local rearrangement. We further introduce a layout solver that fuses semantic scene-graph constraints with visual cues, substantially improving spatial plausibility and global consistency. Extensive experiments show that our method performs superior layout aesthetics and functional realism.

## 1 Introduction

Interactive 3D scene generation has emerged as a rapidly advancing area of research, distinguished by its ability to produce environments that conform to the structural and physical constraints required by modern simulation engines Kolve et al. (2017); Yang et al. (2024c); Yao et al. (2025a); Ling et al. (2025). Unlike generative approaches that directly synthesize 3D scene with diffusion models Fang et al. (2025); Tang et al. (2024); Yang et al. (2024a), interactive scene construction commonly integrates Large Language Models (LLMs) with traditional procedural content generation frameworks to assemble coherent 3D environments Yang et al. (2024c); Zhou et al. (2025). By exploiting pre-existing 3D asset repositories, these approaches provide substantial flexibility and support interactive operation in real time. Consequently, these frameworks are well suited for application across a range of domains, including robotic simulation, immersive virtual environments, and the generation of large-scale synthetic 3D datasets.

Existing methods for interactive 3D indoor scene generation can be broadly categorized into two paradigms. The first involves end-to-end layout synthesis guided by LLMs, in which models are fine-tuned to directly predict numerical layout parameters, such as object positions and orientations Feng et al. (2023); Yang et al. (2025). While effective, these approaches typically depend on large-scale annotated datasets of 3D scenes, limiting their scalability and adaptability. The second paradigm follows an agent-based, modular pipeline, decomposing the task into sequential stages of object recommendation, 3D asset retrieval or generation, and layout refinement via rule- or physics-based optimization Yang et al. (2024c); Çelen et al. (2024); Liu et al. (2025); Zhou et al. (2025). These pipelines are training-free and thus avoid dependence on large datasets. However, they lack the capacity to directly generate globally coherent layouts without resorting to low-level numerical optimizers Zhang et al. (2023); Ling et al. (2025); Yao et al. (2025b). Such optimizers prioritize local geometric constraints but often fail to capture global semantic consistency, resulting in arrangements that are geometrically valid yet contextually implausible. Besides, both paradigms are primarily designed for global layout generation, rendering them ill-suited for incremental object in-

sertion or fine-grained local scene editing. This limitation poses a significant barrier to practical use, where iterative, localized modifications constitute the dominant mode of interaction.

In this paper, we proposes a novel stepwise approach to scene synthesis that addresses fundamental limitations in conventional global placement methodologies. Rather than generating global layouts in a single pass, our framework implements an incremental object placement paradigm that enables adaptive scene construction through sequential additions. Specifically, our approach comprises three stages to construct a 3D scene. First, a room-planning module generate a basic room parameters, e.g. room type, room size, and recommend a collection of 3D asset that should put into the scene. Second, a layout generation module, which uses a Multi-modal Large Language Model (MLLM) to analyzes the current scene state and determine next object's layout. This module producing both approximate spatial coordinates though image prompt techniques Yang et al. (2023) and semantic constraints Yang et al. (2024c). This dual approach ensures that object relationships maintain both spatial coherence and conceptual validity within the generated representations. Third, a layout solver module that transforms both spatial positioning information and semantic relationship constraints into quantitative layout parameters with precise mathematical definitions. This computational approach ensures optimal element placement while maintaining logical hierarchies dictated by content relationships.

The integration of these three modules enables large language models to synthesize indoor environments with enhanced semantic coherence and spatial plausibility. Notably, the stepwise object placement paradigm facilitates straightforward manipulation of scene elements, supporting flexible addition and removal operations mediated through natural language commands. Our approach makes three principal contributions to the field:

- We develop an agentic pipeline that facilitates flexible scene construction and editing through sequential object placement.

- We propose a novel framework that integrates visual information with traditional rule-based constraint solving mechanisms, thereby enabling scene layouts that simultaneously exhibit semantic coherence and spatial plausibility.

- Our experiments demonstrate that stepwise generation methodologies confer significant advantages in terms of scalability and editability.

## 2 RELATED WORK

### 2.1 LLM-BASED GENERATION

Unlike classical rule-based procedural generation (e.g., ProcTHOR Deitke et al. (2022), AI2-THOR Kolve et al. (2017), Infinigen Indoors Raistrick et al. (2024)), LLM-based scene generation either couples an LLM with explicit rules or fine-tunes the LLM end to end. Training-based approaches let the model directly emit layout programs or placement instructions. For example, ReSpace Bucher & Armeni (2025) and LLPlace Yang et al. (2024b) fine-tune an LLM for autoregressive layout generation, which also enables incremental object insertion. OptiScene Yang et al. (2025) synthesizes a large training corpus and combines supervised fine-tuning with DPO Rafailov et al. (2023) so the LLM can produce layouts directly from natural-language instructions. Training-free pipelines typically use an LLM for high-level planning—object selection, scene-graph construction, and constraint formulation—and then solve for metric layouts via optimization. Holodeck improves physical plausibility by predicting spatial relations between assets and solving a constraint optimization problem Yang et al. (2024c). I-Design uses a team of LLM agents to transform text into a scene graph with relative relationships, followed by a placement algorithm to obtain a 2D layout plan Çelen et al. (2024). LayoutGPT retrieves relevant room layouts from a database as in-context exemplars to guide GPT in generating new layouts Feng et al. (2023). These methods rely on high-quality datasets and typically still require robust 3D asset retrieval. Our method is inspired by Holodeck but differs in two key aspects. We integrate visual information and leverage a MLLM to jointly generate semantic constraints and spatial coordinates, ensuring spatially coherent object layouts. Unlike Holodeck's DFS-based search for a globally optimal layout, we adopt a sequential placement strategy, which naturally supports interactive scene editing.

## 2.2 MLLM FOR LAYOUT GENERATION

Recent work shows that MLLMs exhibit strong spatial reasoning. For example, Set-of-Mark prompting improves visual grounding in GPT-4V Yang et al. (2023). Vision-based indoor layout generation similarly benefits from visual cues that encode intuitive object relationships. LayoutVLM leverages an MLLM to predict coarse spatial relations, which are then refined via differentiable optimization to enforce physical plausibility and global consistency Sun et al. (2025). SceneThesis employs an MLLM to derive more reliable scene graphs, producing layouts that better align with input images Ling et al. (2025). Building on these advances, we adopt Set-of-Mark prompting Yang et al. (2023) to guide an MLLM in annotating 2D image coordinates, steering object placement toward plausible configurations.

## 3 METHOD

We present a hierarchical object placement framework, facilitating persistent scene construction and manipulation capabilities. Let $\mathcal{D}$ denote the corpus of available 3D assets. Our methodology necessitates the retrieval of $n$ discrete objects from $\mathcal{D}$ for strategic placement into a scene $S$. Our framework comprises three interconnected modules that collectively enable the systematic construction of 3D indoor scene.

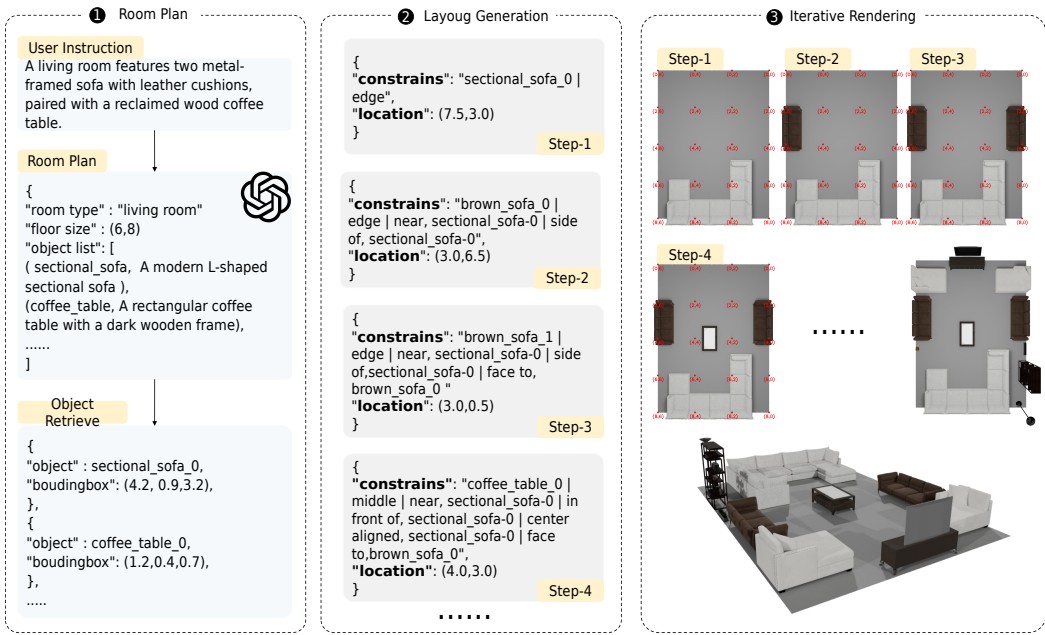

Figure 1: The Architecture of RoomGen. First, we initially employ a LLM to generate fundamental room parameters while concurrently prompting the model to recommend objects for placement within the environment. Second, a MLLM is utilized to interpret the current scene state (top-down view image with red coordinate annotations). Based on this image state, the MLLM generates a recommended coordinate position and spatial constraint. Subsequently, a layout solver algorithm processes these coordinates and spatial constraints to produce precise object positioning coordinates. This process iterates continuously until all objects have been successfully positioned within the scene.

## 3.1 ROOM PLAN MODULE

We implement a room-planning module grounded in a LLM that, given a fixed prompt, maps free-form user instructions to the room's floor-plan dimensions and an associated object inventory. The procedure is entirely prompt-driven, with no task-specific fine-tuning. As illustrated in Figure 1, the

room planning module generates two primary outputs: dimensional parameters defining the room size, and a structured object inventory encoded in JSON format.

After acquisition of the recommended object list, we utilise the object description as a query to retrieve the closest matching object from our pre-indexed 3D asset vector database. The data structure of the acquired objects is illustrated in Figure 1.

## 3.2 Layout Generation Module

After we acquire the object list, we employ a sequential approach to scene composition. To establish precise spatial coordinates, two additional procedural steps are required. First, a MLLM generates approximate positional recommendations and spatial relationship constraints. Subsequently, these parameters are processed through a dedicated layout solver that transforms the qualitative constraints into quantitative spatial coordinates, yielding a numerically precise layout.

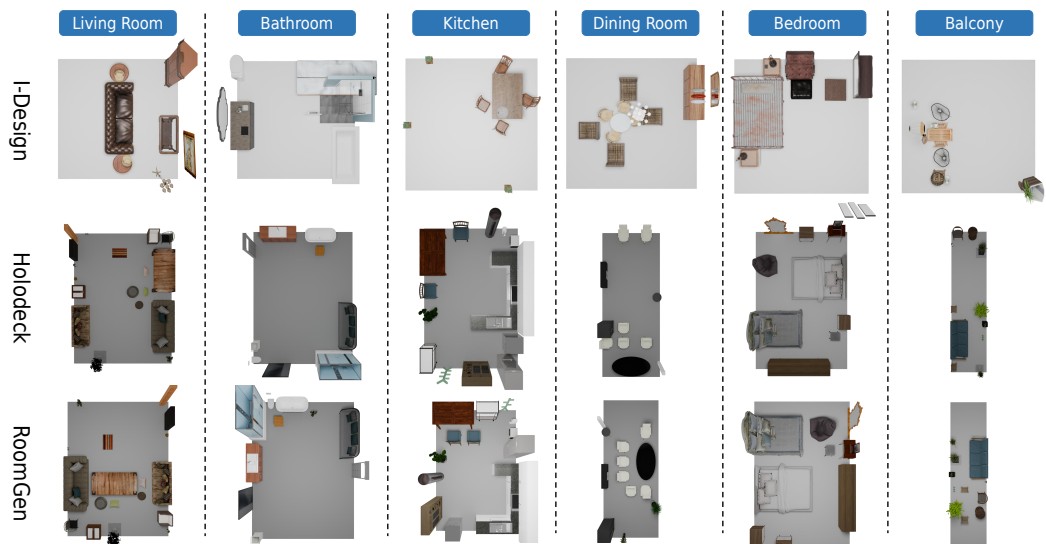

Figure 2: Qualitative comparison of the different layout methods on six types of rooms. I-Design uses a fixed square floor plan, whereas Holodeck and our method enable the LLM to generate free-form floor shapes. Built on Holodeck, our approach uses the same 3D assets across scenes, enabling fair and controlled comparisons of scene layouts.

**Semantic Layout Generation.** In this module, we leverage MLLMs to generate semantic layout representations derived from the current image state. As illustrated in Figure 1, upon placement of each object, we render an image from a top-down perspective, subsequently annotating coordinates to facilitate precise positional determination by the MLLM Yang et al. (2023). At each execution step, the MLLM utilises the rendered image of the existing scene in conjunction with coordinate annotations to recommend spatial coordinates for subsequent object placement, along with the corresponding spatial relationship constraints.

For spatial relationship constraints, we adopt the framework established by Holodeck Yang et al. (2024c), implementing consistent spatial relationship parameters. Specifically, the constraints encompass the following categories: **1. Global Constraint**: edge; middle. **2. Distance Constraint**: near (object); far (object). **3. Position Constraint**: in front of (object); side of (object). **4. Alignment Constraint**: center align with (object). **5. Direction constraint**: face to (object). In practice, we prompt the MLLM to strictly adhere to the constraint types when generating object constraints. The specific prompt design is detailed in the Appendix.

**Numerical Layout Solver.** Based on the established constraints, we identified all viable positions that meet the required criteria by employing algorithms proposed in the Holodeck framework. How-

ever, integration of positions recommended by MLLM necessitated further consideration. Therefore, to comprehensively account for both constraint-satisfying positions and MLLM location recommendations, we develop a novel ranking mechanism that optimizes placement decisions through a systematic prioritization approach.

Through the application of the methodology proposed by Holodeck, we directly obtained candidate layout parameters (position and rotation) satisfying the defined constraints. Each candidate is evaluated based on the number of constraints satisfied, yielding a quantitative score. We establish a ranking system for these positions according to their respective scores, with the resultant ordered set defined as $rank_c$.

Subsequently, we calculated the similarity between each candidate position and the location recommended by the MLLM. These positions are then ranked according to their similarity metrics, with the resulting similarity-based positional ordering defined as $rank_l$.

To integrates scores from different ranking method to generate a composite final score, we design definitive rankings method based on Reciprocal Rank Fusion (RRF) Cormack et al. (2009). Specifically, RRF is defined as: RRF Score $= \sum_{i=1}^{k} \frac{1}{rank_i + c}$. Herein, $rank_i$ represents the ordinal position of a given element within the ith source. $k$ denotes the total number of sources under consideration. $c$ is a constant parameter, conventionally assigned a value of 60. Here we have two ranks source, namely $rank_c$ and $rank_l$. Based on RRF, we define the Layout Score as the follows:

$$\text{Layout Score} = \frac{1}{\text{rank}_c + c} * \alpha + \frac{1}{\text{rank}_l + c} * (1 - \alpha), \tag{1}$$

where $\alpha \in [0, 1]$ governs the relative importance of the position recommended by the MLLM. Using the Layout Score, we balance constraint satisfaction and spatial placement to produce more holistic layouts.

## 4 EXPERIMENTS

### 4.1 SETUP

In all experiments, we use the *GPT-4o-2024-11-20* [1] model for both experimentation and evaluation. We evaluate our method on six room categories: balcony, bathroom, bedroom, dining room, kitchen, and living room. For each category, we generate 25 scenes to compute aggregate statistics.

For 3D assets retrieval, we use assets from the HSSD dataset Khanna et al. (2024). Each assets are annotated by *Qwen2.5-VL-72B* [2] models using four viewpoint images to produce precise object descriptions. We encode object description with *CLIP-ViT-B-32-Text* encoder and images with *CLIP-ViT-B-32-Image* encoder, yielding one text embedding and four view-specific image embeddings per object [3]. For the Layout Score, we set the $\alpha = 0.5$ for default method.

### 4.2 EVALUATION METRICS

For evaluation, we adopted a multi-criteria framework. We evaluate the consistency between design prompts and generated scenes by CLIP similarity. Furthermore, we employ MLLM to evaluate scene from four different perspectives. 1. *Object Pose (OP)*: Assesses positional accuracy, orientation rationality, proportional relationships, and spatial distances between objects (e.g., functional alignment, realistic size ratios, appropriate gaps). 2. *Semantic Consistency (SC)*: Evaluates logical matching between objects and scene type, and between objects themselves (e.g., functional relevance, scenario appropriateness). 3. *Scene Functionality (SF)*: Measures practical usability via traffic flow smoothness, functional zoning clarity, ergonomic spacing, and space utilization efficiency. 4. *Visual Aesthetics (VA)*: Assesses spatial balance, stylistic coherence, and arrangement orderliness.

---

[1] https://platform.openai.com/docs/models

[2] https://huggingface.co/Qwen

[3] https://huggingface.co/openai/clip-vit-base-patch32

Table 1: Main Result. We systematically benchmark competing methods across six room categories, and evaluate performance with five metrics: CLIP similarity (CLIP-Sim), Object Pose (OP), Semantic Consistency (SC), Scene Functionality (SF), and Visual Aesthetics (VA).

| | Method | ↑ CLIP | ↑ OP | ↑ SC | ↑ SF | ↑ VA |
|---|---|---|---|---|---|---|
| Living Room | Idesign | $0.18_{\pm 0.02}$ | $7.75_{\pm 1.23}$ | $8.45_{\pm 1.49}$ | $6.71_{\pm 1.27}$ | $7.58_{\pm 1.52}$ |
| | Holodeck | $0.19_{\pm 0.02}$ | $7.48_{\pm 0.57}$ | $8.16_{\pm 1.15}$ | $6.40_{\pm 0.69}$ | $7.28_{\pm 0.91}$ |
| | Ours | $\mathbf{0.22}_{\pm 0.01}$ | $\mathbf{7.92}_{\pm 0.39}$ | $\mathbf{8.80}_{\pm 0.49}$ | $\mathbf{6.76}_{\pm 0.51}$ | $\mathbf{7.80}_{\pm 0.49}$ |
| Bathroom | Idesign | $0.21_{\pm 0.03}$ | $6.66_{\pm 0.94}$ | $7.50_{\pm 1.75}$ | $5.41_{\pm 1.25}$ | $6.25_{\pm 1.23}$ |
| | Holodeck | $0.23_{\pm 0.02}$ | $7.64_{\pm 0.74}$ | $7.96_{\pm 1.37}$ | $6.44_{\pm 0.94}$ | $7.32_{\pm 1.19}$ |
| | Ours | $\mathbf{0.24}_{\pm 0.01}$ | $\mathbf{8.24}_{\pm 0.41}$ | $\mathbf{9.01}_{\pm 0.69}$ | $\mathbf{7.08}_{\pm 0.68}$ | $\mathbf{8.28}_{\pm 0.45}$ |
| Kitchen | Idesign | $0.21_{\pm 0.02}$ | $7.74_{\pm 0.72}$ | $8.41_{\pm 1.38}$ | $6.75_{\pm 0.72}$ | $7.83_{\pm 0.68}$ |
| | Holodeck | $0.21_{\pm 0.02}$ | $7.68_{\pm 0.67}$ | $7.96_{\pm 1.31}$ | $6.44_{\pm 0.75}$ | $7.08_{\pm 1.23}$ |
| | Ours | $\mathbf{0.24}_{\pm 0.01}$ | $\mathbf{8.24}_{\pm 0.43}$ | $\mathbf{9.24}_{\pm 0.42}$ | $\mathbf{7.24}_{\pm 0.42}$ | $\mathbf{8.28}_{\pm 0.44}$ |
| Dining Room | Idesign | $0.18_{\pm 0.03}$ | $\mathbf{8.13}_{\pm 0.62}$ | $\mathbf{8.91}_{\pm 0.91}$ | $\mathbf{7.08}_{\pm 0.74}$ | $8.11_{\pm 0.69}$ |
| | Holodeck | $0.19_{\pm 0.03}$ | $7.74_{\pm 0.63}$ | $7.88_{\pm 1.11}$ | $6.36_{\pm 0.74}$ | $7.16_{\pm 1.04}$ |
| | Ours | $\mathbf{0.21}_{\pm 0.02}$ | $7.92_{\pm 0.62}$ | $8.6_{\pm 0.56}$ | $6.84_{\pm 0.67}$ | $\mathbf{7.84}_{\pm 0.67}$ |
| Bedroom | Idesign | $0.21_{\pm 0.02}$ | $7.55_{\pm 0.76}$ | $8.13_{\pm 1.30}$ | $6.41_{\pm 0.92}$ | $7.24_{\pm 1.24}$ |
| | Holodeck | $0.22_{\pm 0.02}$ | $7.76_{\pm 0.86}$ | $8.68_{\pm 1.01}$ | $6.72_{\pm 0.91}$ | $7.60_{\pm 1.21}$ |
| | Ours | $\mathbf{0.24}_{\pm 0.02}$ | $\mathbf{8.24}_{\pm 0.24}$ | $\mathbf{9.16}_{\pm 0.33}$ | $\mathbf{7.24}_{\pm 0.42}$ | $\mathbf{8.28}_{\pm 0.45}$ |
| Balcony | Idesign | $0.17_{\pm 0.02}$ | $7.22_{\pm 0.97}$ | $7.44_{\pm 1.70}$ | $6.01_{\pm 1.24}$ | $7.00_{\pm 1.24}$ |
| | Holodeck | $0.21_{\pm 0.03}$ | $7.56_{\pm 0.57}$ | $8.44_{\pm 1.02}$ | $6.48_{\pm 0.85}$ | $7.68_{\pm 0.83}$ |
| | Ours | $\mathbf{0.24}_{\pm 0.01}$ | $\mathbf{8.28}_{\pm 0.45}$ | $\mathbf{9.08}_{\pm 0.27}$ | $\mathbf{7.16}_{\pm 0.36}$ | $\mathbf{8.40}_{\pm 0.49}$ |

## 4.3 MAIN RESULT

To evaluate the efficacy of our approach in scene generation, we conducted comparative analyses against two leading text-to-scene methodologies, Holodeck and I-Design. As illustrated in Table 1, our method demonstrates superior performance across nearly all metrics and room typologies, underscoring its robust generalizability and algorithmic advancement in spatial synthesis tasks. It is important to note that our methodology relies on Holodeck for both constraint specification and constraint-based layout solving. Nevertheless, our approach demonstrates superior performance compared to Holodeck, thereby validating the efficacy of vision-based layout generation approaches

Moreover, we observed that both our approach and Holodeck demonstrated inferior performance compared to I-Design in the Dining Room scenario. We observe that I-desin typically generates scenes with chairs arranged around a table, regardless of their orientations (seeing Figure 2). In contrast, Holodeck and our method exhibit greater scene diversity—chairs are not necessarily placed around the table. This difference primarily accounts for the lower scores on the evaluation metrics.

As shown in Figure 2, we provide qualitative comparisons across six room categories. First, we observe that I-Design consistently produces scenes with fewer objects. As the object count increases, I-Design exhibits a high failure rate and incurs substantial LLM token costs Çelen et al. (2024). In contrast, our method can, in principle, populate scenes with arbitrarily many objects, while its LLM token usage grows only linearly with the number of objects. Second, our method better respects functional constraints of objects compared to Holodeck. In the living room case, Holodeck often places the coffee table against a wall, whereas we place it between two sofas. In the kitchen case, Holodeck positions the refrigerator in a corner facing a cabinet, which prevents the door from opening. In the dining room case, Holodeck struggles with an excess of chairs and places several against the wall. In comparison, our method demonstrates superior results in layout plausibility and object functionality compared to existing methods.

## 4.4 ABLATION

### 4.4.1 THE IMPORTANT OF VISION INFORMATION

**Method Settings.** Our methodology primarily leverages MLLM integrated with visual information to generate constrains and location, whereas our baseline models such as Holodeck operate

Table 2: Ablation Study. We assess performance across three settings, each controlling the proportion of visual information.

| | Method | ↑ CLIP | ↑ OP | ↑ SC | ↑ SF | ↑ VA |
|---|---|---|---|---|---|---|
| Living Room | Only Constrains | $0.19_{\pm 0.02}$ | $7.44_{\pm 0.57}$ | $8.36_{\pm 0.74}$ | $6.52_{\pm 0.64}$ | $7.32_{\pm 0.78}$ |
| | Image-based Constrain | $0.21_{\pm 0.02}$ | $7.76_{\pm 0.58}$ | $8.76_{\pm 0.58}$ | $6.76_{\pm 0.58}$ | $7.72_{\pm 0.66}$ |
| | Constraints + Location ($\alpha = 0.3$) | $0.21_{\pm 0.01}$ | $7.61_{\pm 0.48}$ | $8.44_{\pm 0.75}$ | $6.52_{\pm 0.57}$ | $7.52_{\pm 0.64}$ |
| | Constraints + Location ($\alpha = 0.5$) | $\mathbf{0.22}_{\pm 0.01}$ | $\mathbf{7.92}_{\pm 0.39}$ | $\mathbf{8.80}_{\pm 0.49}$ | $\mathbf{6.76}_{\pm 0.51}$ | $\mathbf{7.80}_{\pm 0.49}$ |
| Bathroom | Only Constrains | $0.22_{\pm 0.02}$ | $7.60_{\pm 0.74}$ | $8.20_{\pm 1.23}$ | $6.48_{\pm 0.85}$ | $7.36_{\pm 1.16}$ |
| | Image-based Constrain | $0.22_{\pm 0.02}$ | $7.96_{\pm 0.66}$ | $8.24_{\pm 1.27}$ | $6.76_{\pm 0.71}$ | $7.88_{\pm 0.71}$ |
| | Constraints + Location ($\alpha = 0.3$) | $0.22_{\pm 0.01}$ | $7.76_{\pm 0.94}$ | $8.12_{\pm 1.33}$ | $6.60_{\pm 0.84}$ | $7.80_{\pm 0.49}$ |
| | Constraints + Location ($\alpha = 0.5$) | $\mathbf{0.24}_{\pm 0.01}$ | $\mathbf{8.24}_{\pm 0.41}$ | $\mathbf{9.01}_{\pm 0.69}$ | $\mathbf{7.08}_{\pm 0.68}$ | $\mathbf{8.28}_{\pm 0.45}$ |
| Kitchen | Only Constrains | $0.23_{\pm 0.01}$ | $7.60_{\pm 0.69}$ | $8.20_{\pm 0.97}$ | $6.48_{\pm 0.75}$ | $7.44_{\pm 0.80}$ |
| | Image-based Constrain | $0.22_{\pm 0.02}$ | $7.76_{\pm 0.65}$ | $8.40_{\pm 0.93}$ | $6.64_{\pm 0.68}$ | $7.64_{\pm 0.68}$ |
| | Constraints + Location ($\alpha = 0.3$) | $0.22_{\pm 0.03}$ | $7.84_{\pm 0.88}$ | $8.64_{\pm 0.93}$ | $6.76_{\pm 0.64}$ | $7.80_{\pm 0.49}$ |
| | Constraints + Location ($\alpha = 0.5$) | $\mathbf{0.24}_{\pm 0.01}$ | $\mathbf{8.24}_{\pm 0.43}$ | $\mathbf{9.24}_{\pm 0.42}$ | $\mathbf{7.24}_{\pm 0.42}$ | $\mathbf{8.28}_{\pm 0.44}$ |
| Dining Room | Only Constrains | $0.18_{\pm 0.03}$ | $7.48_{\pm 0.57}$ | $8.08_{\pm 1.01}$ | $6.36_{\pm 0.68}$ | $7.32_{\pm 0.83}$ |
| | Image-based Constrain | $0.20_{\pm 0.02}$ | $7.76_{\pm 0.64}$ | $8.28_{\pm 0.96}$ | $6.68_{\pm 0.73}$ | $7.68_{\pm 0.78}$ |
| | Constraints + Location ($\alpha = 0.3$) | $0.20_{\pm 0.02}$ | $7.76_{\pm 0.76}$ | $8.56_{\pm 0.81}$ | $6.72_{\pm 0.60}$ | $7.80_{\pm 0.49}$ |
| | Constraints + Location ($\alpha = 0.5$) | $\mathbf{0.21}_{\pm 0.02}$ | $\mathbf{7.92}_{\pm 0.62}$ | $\mathbf{8.60}_{\pm 0.56}$ | $\mathbf{6.84}_{\pm 0.67}$ | $\mathbf{7.84}_{\pm 0.67}$ |
| Bedroom | Only Constrains | $0.21_{\pm 0.02}$ | $7.84_{\pm 0.67}$ | $8.76_{\pm 0.86}$ | $6.80_{\pm 0.74}$ | $7.68_{\pm 0.92}$ |
| | Image-based Constrain | $0.23_{\pm 0.02}$ | $8.12_{\pm 0.71}$ | $8.88_{\pm 0.91}$ | $7.01_{\pm 0.74}$ | $8.01_{\pm 0.74}$ |
| | Constraints + Location ($\alpha = 0.3$) | $0.24_{\pm 0.02}$ | $8.08_{\pm 0.62}$ | $8.92_{\pm 0.84}$ | $7.04_{\pm 0.59}$ | $8.04_{\pm 0.59}$ |
| | Constraints + Location ($\alpha = 0.5$) | $\mathbf{0.24}_{\pm 0.02}$ | $\mathbf{8.24}_{\pm 0.24}$ | $\mathbf{9.16}_{\pm 0.33}$ | $\mathbf{7.24}_{\pm 0.42}$ | $\mathbf{8.28}_{\pm 0.45}$ |
| Balcony | Only Constrains | $0.22_{\pm 0.02}$ | $8.04_{\pm 0.52}$ | $8.88_{\pm 0.76}$ | $6.96_{\pm 0.59}$ | $8.08_{\pm 0.68}$ |
| | Image-based Constrain | $0.23_{\pm 0.02}$ | $8.16_{\pm 0.54}$ | $8.92_{\pm 0.79}$ | $7.08_{\pm 0.48}$ | $8.32_{\pm 0.46}$ |
| | Constraints + Location ($\alpha = 0.3$) | $0.23_{\pm 0.02}$ | $8.04_{\pm 0.52}$ | $8.80_{\pm 0.63}$ | $7.00_{\pm 0.56}$ | $8.36_{\pm 0.55}$ |
| | Constraints + Location ($\alpha = 0.5$) | $\mathbf{0.24}_{\pm 0.01}$ | $\mathbf{8.28}_{\pm 0.45}$ | $\mathbf{9.08}_{\pm 0.27}$ | $\mathbf{7.16}_{\pm 0.36}$ | $\mathbf{8.40}_{\pm 0.49}$ |

without visual input. We aims to investigate the significance of visual information in our method. To this end, we evaluate three conditions with varying degrees of visual information utilization: 1. **Only Constraints**: Under this setting, we exclusively employ constraints for sequential scene generation. The process utilized a LLM to generate constraints for subsequent objects based on the current constraint state, without incorporating any visual information. And the layout solver module only use the constraint to generate the layout parameters. 2. **Image-based Constraints**: This approach similar the 'Only Constraints' setting, with the critical distinction being the implementation of a MLLM. The current scene rendering image and current constraint state are provided to the MLLM, which then generated constraints for the next object. 3. **Constraints + Location** ($\alpha = 0.3$): Building upon our standard methodology, we modulated the $\alpha$ to 0.3. This setting biases the Layout Score toward the location cues predicted by the MLLM. Please note that the $\alpha$ is set to 0.5 by default. Therefore, the Constraints + Location ($\alpha = 0.5$) is our default method, we copy the result from Table 1 to Table 2 for convenient. Moreover, the Image-based Constraints configuration is equivalent to setting $\alpha = 1.0$.

**Comparation Results.** As illustrated in Table 2, we evaluate the ablation method using six types of room and five evaluation metrics. We observe that employing solely LLMs for constraint generation yields the least performance. In contrast, 'Image-based Constrain' setting demonstrates performance enhancements across virtually all room configurations and evaluation metrics. These results provide compelling evidence for the critical importance of visual information in such applications.

Furthermore, we adjust the proportion of visual information (location) in the Reciprocal Rank Fusion (RRF) framework, increasing the importance of visual signals. However, we observe no significant performance improvement under $\alpha = 0.3$ setting. This finding underscores the critical role of spatial constraints in relationship modeling. The main reason is that our method primarily utilises constraints to generate candidate positions, which inherently emphasises their importance. However, 'Image-based Constrain', which relies solely on constraint signals to produce the final layout, performs worse than 'Constraints + Location ($\alpha = 0.3$)' and 'Constraints + Location ($\alpha = 0.5$)'. This further demonstrates the importance of the MLLM-generated location cues.

Table 3: The percentage of finally solved layout aligns with the MMLM's location predictions.

| Method | Living Room | Bathroom | Kitchen | Dining Room | Bedroom | Balcony |
|---|---|---|---|---|---|---|
| Constraints + Location ($\alpha = 0.5$) | 7.73 | 12.56 | 9.52 | 8.61 | 11.80 | 19.47 |
| Constraints + Location ($\alpha = 0.3$) | 8.67 | 13.63 | 10.04 | 12.50 | 14.66 | 20.55 |

### 4.5 FINAL OBJECT PLACEMENT VS MLLM-SUGGESTED LOCATION

To assess the role of visual information in our system, we quantify the spatial discrepancy between the MLLM-suggested placement and the final object placement. Specifically, we report the fraction of trials where the Euclidean distance between these two locations is below 1 meter.

Surprisingly, as showed in Table 3, this proportion was approximately 10%, and did not increase significantly when the distance threshold is raised. This proportion is substantially lower than expected. This finding suggests substantial incongruity between MLLM-suggested positions and constraint-driven positions. However, despite this discrepancy, the overall scene generation performance is enhanced. Given that our approach is training-free, it relies entirely on the MLLM's inherent spatial reasoning capabilities. We posit that achieving greater alignment between MLLM-generated location and constraint-based recommendations would necessitate further fine-tuning of the model architecture.

### 4.6 THE ADVANTAGE OF STEP-BY-STEP GENERATION

Our methodology employs a stepwise generation approach, offering enhanced flexibility compared to global optimization techniques. However, this approach presents certain limitations that warrant consideration.

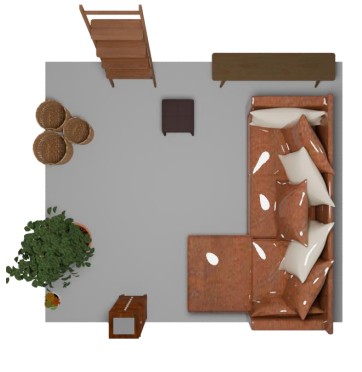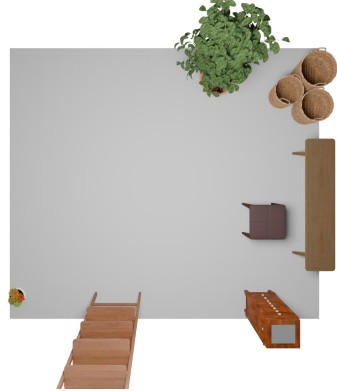

Figure 3: Global Layout Planning (Left) vs. Stepwise Layout (Right)

Here we compare with the global layout solver method (Holodeck) and our stepwise method. In situations where objects collectively occupy nearly the entire available space, global optimization techniques can efficiently arrange all items to maximize spatial utilization. The stepwise approach, however, may encounter limitations wherein the initial placement of objects potentially restricts the available space for subsequent items, resulting in suboptimal final configurations. As illustrated in Figure 3, the Stepwise method terminates once it detects that the sofa cannot fit into the designated space. We think that this represents a fundamental trade-off between methodologies, with each approach offering distinct advantages in specific scenarios. This balance between local flexibility and global optimization remains an important consideration in spatial arrangement systems.

Despite their limitations in the aforementioned setting, stepwise methods enable incremental object insertion, making scene editing feasible and facilitating post-hoc optimization. As shown in Figure 9, our method supports incremental object insertion, enabling flexible scene editing and facilitating deployment in real-world scenarios.

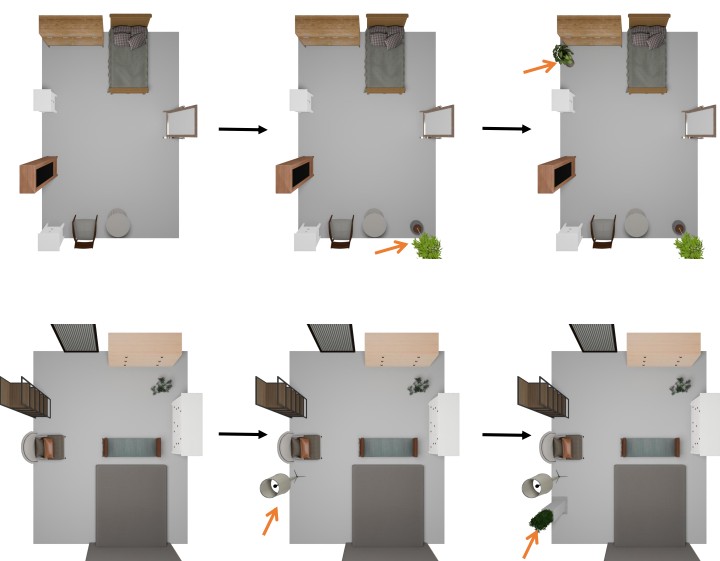

Figure 4: Incremental Object Insertion

## 5 CONCLUSION

In this paper, we presents a stepwise framework for indoor scene generation. We leverage a multi-modal large language model (MLLM) to iteratively place objects in the scene, enabling flexible scene synthesis and editing. We integrate rule-based constraints with a MLLM for spatial localization, producing scenes that are both spatially and functionally plausible. Moreover, our stepwise pipeline reduces scene construction to linear-time complexity and enables flexible scene editing. We conduct comprehensive comparisons against strong baselines, demonstrating clear advantages of our approach. However, our method is entirely train-free, thus its upper bound of performance is constrained by the capacity of the underlying MLLMs. Nevertheless, we argue that step-wise strategies will become increasingly important for scene generation and editing. In future work, we will transition from train-free to train-based variants to further improve performance and robustness in 3D indoor scenes.

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

# A APPENDIX

## A.1 LLM USAGE DECLARATION

We only use LLMs for language polishing in the process of writing papers.

## A.2 PROMPT

```
Room Plan Prompt:
## Roles
You are an experienced room designer, please assist me in planning a room size
and selecting floor objects to furnish the room. You need to select appropriate
objects to satisfy the customer's requirements. You must provide a description and
desired size for each object since I will use it to retrieve object. If multiple
items are to be placed in the room with the same description, please indicate
the quantity and variance_type ("same" if they should be identical, otherwise
"varied").

## Response Format
{ "room_size": [x,y],
"object_name":{ "description": a short sentence describing the object,
"size": the desired size of the object, in the format of a list of three numbers,
[length, width, height] in centimeters,
"quantity": the number of objects (int),
"variance_type": "same" or "varied", }
}

## Guidelines
Here are some guidelines for you:
1. Provide reasonable room size based on the number of objects in the request.
2. Do not provide rug/mat, windows, doors, curtains, and ceiling objects which
have been installed for each room.
3. A room's size range (length or width) is 3m to 20m. The maximum area of a room
is 48 m². Please provide a floor plan within this range and ensure the room is not
too small or too large.
4. I want more types of large objects and more types of small objects on top of
the large objects to make the room look more vivid.
5. Make sure the number of objects can almost fit in the room according to the
room size (at least 5-10 objects).

## User's Request
Currently, the user request is {input}.

Your response should be direct and without additional text at the beginning or end.
```

Figure 5: Prompt Templates for Room Plan.

**Constrain and Location Generation Prompt:**
## Roles
You are an indoor object placement expert who excels at planning layouts that are both aesthetically pleasing and functionally look. I will provide you with an image showing the current layout, you need to help me decide where to place the next object. The image has marked coordinates, please refer to this coordinate system. Besides, please help me arrange objects in the room by assigning constraints to each object.
## Coordinate System
The coordinate system in the image works as follows:
- The origin **(0,0)** is at the **top-right** corner.
- The **x-values increase downward**.
- The **y-values increase to the left**.
## Constrains
Here are the constraints and their definitions:
1. global constraint:
1) edge: at the edge of the room, close to the wall, most of the objects are placed here.
2) middle: not close to the edge of the room.
2. distance constraint:
1) near, object: near to the other object, but with some distance, 50cm < distance < 150cm.
2) far, object: far away from the other object, distance >= 150cm.
3. position constraint:
1) in front of, object: in front of another object.
2) around, object: around another object, usually used for chairs.
3) side of, object: on the side (left or right) of another object.
4) left of, object: to the left of another object.
5) right of, object: to the right of another object.
4. alignment constraint:
1) center aligned, object: align the center of the object with the center of another object.
5. Rotation constraint:
1) face to, object: face to the center of another object.
For each object, you must have one global constraint and you can select various numbers of constraints and any combinations of them and the output format must be:
object | global constraint | constraint 1 | constraint 2 | ...

For example:
sofa-0 | edge
coffee table-0 | middle | near, sofa-0 | in front of, sofa-0 | center aligned, sofa-0 | face to, sofa-0
tv stand-0 | edge | far, coffee table-0 | in front of, coffee table-0 | center aligned, coffee table-0 | face to, coffee table-0 desk-0 | edge | far, tv stand-0
chair-0 | middle | in front of, desk-0 | near, desk-0 | center aligned, desk-0 | face to, desk-0
floot lamp-0 | middle | near, chair-0 | side of, chair-0

## Guidelines
### For constrains
1. I will use your guideline to arrange the objects *iteratively*, so please start with an anchor object which doesn't depend on the other objects (with only one global constraint).
2. Place the larger objects first.
3. The latter objects could only depend on the former objects.
4. The objects of the *same type* are usually *aligned*.
5. I prefer objects to be placed at the edge (the most important constraint) of the room if possible which makes the room look more spacious.
6. When handling chairs, you should use the around position constraint. Chairs must be placed near to the table/desk and face to the table/desk.

### For location
1. Do not give the location out of the floor boundary.
2. The coordinates shown in the image are just grid intervals, you can provide any coordinates (including decimal values).
3. Give the reasoning of why this locaition is reasonable.
4. When selecting an object, primarily consider its relationship to the existing objects and ensure the placement matches the object's functional characteristics.

## Response Format
{ "reasoning": "reasoning of why this locaition is reasonable",
"constrains": "constrains of the object"
"location": [x, y]
}
## Current Task
The floor boundary is {axis_bound}

The placed objects in the scene are {placed_objects}

The current constrains are {constrains}

The next object you should place is {object_name}.

Please first use natural language to explain your high-level design strategy, and then follow the desired format *strictly* (do not add any additional text at the beginning or end) to provide the constraints for each object.

Figure 6: Prompt Templates for Constrain and Location Generation

**MLLM Evaluation Prompts:**
## Role
You are a professional evaluator specializing in indoor functional logic, ergonomics, and aesthetic design, tasked with objective, evidence-based assessment of top-down indoor layouts across 5 core dimensions, strictly adhering to "Object Pose, Physical Reality, Semantic Consistency, Scene Functionality, and Visual Aesthetics" principles. Evaluations must rely exclusively on provided text descriptions and top-down images, with no subjective inferences about unmentioned details.

## Core Evaluation Dimensions
1. **Object Pose**: Assesses positional accuracy, orientation rationality, proportional relationships, and spatial distances between objects (e.g., functional alignment, realistic size ratios, appropriate gaps).
2. **Semantic Consistency**: Evaluates logical matching between objects and scene type, and between objects themselves (e.g., functional relevance, scenario appropriateness).
3. **Scene Functionality**: Measures practical usability via traffic flow smoothness, functional zoning clarity, ergonomic spacing, and space utilization efficiency.
4. **Visual Aesthetics**: Assesses spatial balance, stylistic coherence, and arrangement orderliness.

## Evaluation Rules
- **Information Boundary**: Limited to "scene description" and 1 top-down renderings. Note "insufficient image details" for ambiguous elements.
- **Scoring (0-10)**: 10=perfect; 8-9=excellent (negligible flaws); 6-7=good (minor issues); 4-5=partial compliance (obvious defects); 2-3=poor (major flaws); 0-1=non-compliant (invalid layout).
- **Scope**: Comprehensive 5-dimensional assessment of the entire scene.

## Scene Information
- **Description**: the user's request is : {prompt};
- **Images**: 1 top-down renderings (full scene coverage, no blind spots).

## Response Format
Standard JSON with scores (0-10) and evidence-based comments (linking text and image details) for each dimension: {
"Object Pose": {"Score": 8, "Comment": "Consistent with scene description stating 'dining chairs arranged around table'|images show 4 chairs aligned with table edges (65cm spacing, consistent with ergonomic standards). Minor deviation in one chair's orientation (5° off) does not affect functionality."},
"Semantic Consistency": {"Score": 6, "Comment": "Most objects match 'living room' description (sofa, TV, coffee table) per images, but text-specified 'bookshelf' is absent, creating a minor semantic gap."},
"Scene Functionality": {"Score": 7, "Comment": "Main passage (100cm) meets standards (90cm) as shown in images, aligning with text's 'smooth traffic flow' claim. Minor crowding in corner (20cm gap between cabinet and sofa) reduces efficiency." },
"Visual Aesthetics": {"Score": 9, "Comment": "Images show balanced spatial distribution (no weight bias) and unified modern style, consistent with text's 'neat arrangement' description. Minimal asymmetry in decor placement is negligible." }
}

Figure 7: Prompt Templates for MLLM Evaluation

## A.3 STEPWISE LAYOUT GENERATION EXAMPLE

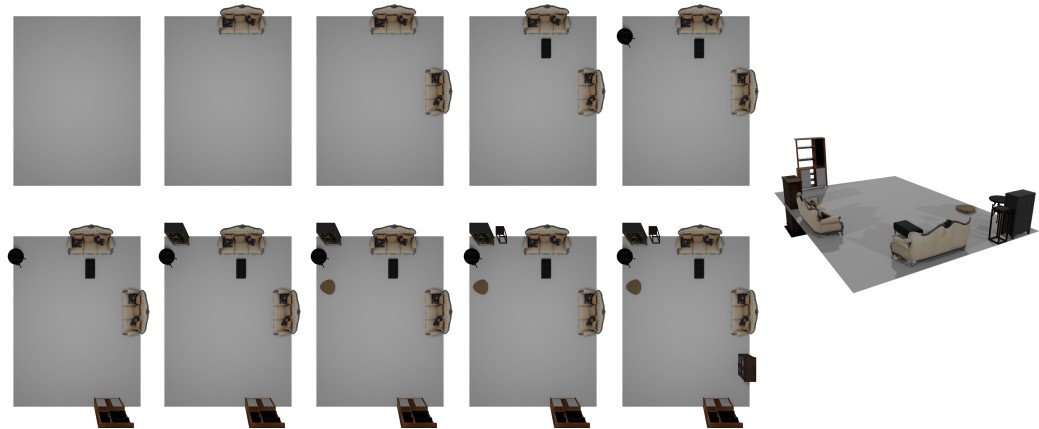

Figure 8: A living room example.

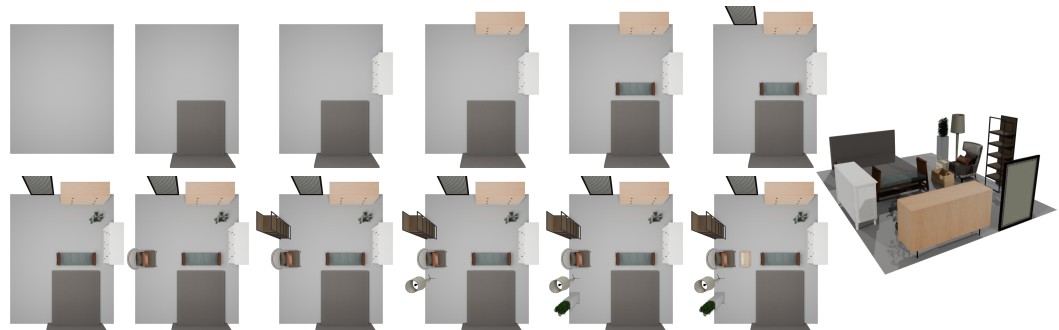

Figure 9: A bedroom example.

