# OpenReview forum: "RoomGen: Text-Driven Agentic 3D In-door Scene Synthesis and Editing"
_ICLR.cc/2026/Conference — Submitted to ICLR 2026_

### Official Review · Reviewer_QNT4 · 2025-10-30

**Soundness:** 2
**Presentation:** 2
**Contribution:** 1
**Rating:** 2
**Confidence:** 4

**Summary:**

The paper introduces RoomGen, an MLLM-based framework for 3D indoor scene generation. Unlike global scene generation methods that do not support incremental editing, RoomGen generates scenes sequentially by placing objects conditioned on previously placed ones. It also integrates visual cues via MLLMs to promote semantic coherence and spatial plausibility.

RoomGen first generates room parameters and recommends a set of relevant 3D assets using an LLM. It then recommends spatial coordinates and semantic constraints for the next object conditioned on the current scene state. Finally, a numerical layout solver processes these coordinates and constraints to produce the object’s final placement.

**Strengths:**

1) The proposed pipeline attempts to integrate visual cues into constraint solving mechanisms to ensure semantic coherence and spatial plausibility.
2) Sequential object placement can be feasible for some aspects of scene editing.
3) The paper is written in an easy-to-follow manner.

**Weaknesses:**

1) Lack of global re-optimization/correction: The paper claims that sequential object placement benefits flexible placement and incremental editing. However, Section 4.6 shows why an approach without global correction/re-optimization is not directly suitable for incremental editing (L422–424: “...the initial placement of objects potentially restricts the available space for subsequent items...” ). A global layout solver or feedback mechanism can be applied before each newly inserted object to mitigate such limitations and achieve the best of both worlds.
2) Limited novelty: The paper is built on Holodeck and relies on off-the-shelf LLMs/MLLMs across multiple stages, yielding good scene generation results. However, despite introducing a sequential generation and using visual cues, the framework offers somewhat limited technological innovation. As the basic limitation of sequential generation is not addressed, the contributions of two claimed innovations are less convincing.
3) Lack of comparison with recent baselines after Holodeck (e.g., LayoutVLM [1])
4) Generation time: The approach queries the MLLM multiple times across stages of sequential generation, which likely makes it slower than global methods. However, generation time is not reported.

[1] Sun, F. Y., Liu, W., Gu, S., Lim, D., Bhat, G., Tombari, F., ... & Wu, J. (2025). Layoutvlm: Differentiable optimization of 3d layout via vision-language models. In Proceedings of the Computer Vision and Pattern Recognition Conference (pp. 29469-29478).

**Questions:**

1) L164-166: How does object retrieval work? Does the framework also compute Euclidean distance between the predicted bounding boxes and the asset bounding boxes in the dataset, in addition to using the descriptions?
2) Table 3: When alpha=0.3, spatial coordinates predicted by MLLM becomes more important and this setup seems to give a higher percentage of final layouts aligned with the MMLM’s predictions. Does this imply that adding semantic constraints actually degrades the MLLM’s recommendations? An ablation study is presented in Section 4.4.1, but it is not designed to prove the importance of constraints.

---

### Official Review · Reviewer_4jZA · 2025-10-31

**Soundness:** 2
**Presentation:** 2
**Contribution:** 1
**Rating:** 2
**Confidence:** 4

**Summary:**

This paper proposes a 3-step method to generate a room layout. It starts with a room plan powered by off-the-shelf LLM. It then iteratively adds objects into the scene by combining an off-the-shelf LLM, which suggests the location of the newly added object, and a heuristic-based solver, which takes both the suggested location and the room constraints into consideration. Experiments are conducted on 6 types of rooms with 2 baselines, and numbers are reported with automatic evaluations.

**Strengths:**

1. The proposed method shows some improvements compared to the Holodeck baseline, at least under the evaluation metrics the authors use.
2. The paper shows a potential way to generate the location of new objects given an existing room configuration, which could be a useful task.

**Weaknesses:**

- Problem with Evaluation: The main evaluation is based on CLIP scores and several metrics judged by VLMs. There are sever problems with this:
1) It is unclear what is the image input to the CLIP model. I assume it is the birds-eye-view (BEV) of the scene. Visually, the rendered BEV images diverge from the natural images CLIP was trained on. The text input to CLIP is also a long sentence prompt, which CLIP is not designed to handle. Therefore, it is unclear if CLIP is good for such evaluation.
2) The VLM evaluation (Object Pose (OP), Semantic Consistency (SC), Scene Functionality (SF), Visual Aesthetics (VA)) is not standard. It is also unclear if general-purpose VLM can be used as judge for the room layout generation, and how well does the results align with human judgements.
3) Room layout has a high degree of freedom and is very hard to judge automatically. In this paper, no human evaluation provided, and the qualitative results are very limited.
4) Even out of the limited qualitative results, results are not promising. Fig 8 has an unproportionally small table in front of the sofa, and the room is very empty. Fig 9 the bed is blocked by other furnitures and is not accessible.

- Missing baselines: Direct Numerical Layout Generation for 3D Indoor Scene Synthesis via Spatial Reasoning, Neurips2025

- Limited Novelty and Contribution: the authors simply adopted off-the-shelf GPT as the agent, and the way to prompt it is standard. One contribution the authors claim, is the Numerical Layout Solver, which is basically the weighted average of two rankings (constraints-based ranking and VLM-based ranking). This design seems ad hoc, and it doesn't work: in sec4.5, the authors report that the VLM's suggestion is only respected 10% of the time.

- Limitation due to BEV representation: the proposed solution cannot handle complex object relationships, such as objects put on the top of a table or on a multi-level shelf. This is not a problem for the Direct Numerical Layout Generation for 3D Indoor Scene Synthesis via Spatial Reasoning baseline.

- (Minor) Typos: Fig1 "Layoug Generation" -> "Layout Generation"

**Questions:**

N/A

---

### Official Review · Reviewer_KDjv · 2025-11-01

**Soundness:** 2
**Presentation:** 2
**Contribution:** 2
**Rating:** 2
**Confidence:** 4

**Summary:**

This paper presents RoomGen, a text-driven, agent-based framework for interactive 3D indoor scene generation and editing.

Unlike prior “retrieve-then-place” or global optimization systems (e.g., Holodeck, I-Design), RoomGen proposes a stepwise object placement paradigm that incrementally constructs scenes using three modules:

1. Room Planning Module: an LLM-based planner that infers room size and retrieves a relevant list of 3D assets from a database;
2. Layout Generation Module: a multimodal LLM (MLLM) that interprets the current scene (via top-down renderings) and predicts spatial constraints and approximate object positions;
3. Layout Solver Module: fuses semantic constraints with visual recommendations through Reciprocal Rank Fusion (RRF) to determine final object placements.

The paper claims three main contributions:
1. A sequential agentic pipeline enabling flexible scene generation and editing;
2. Integration of visual grounding (via MLLM) with rule-based constraint solving for improved spatial plausibility;
3. Demonstration that stepwise generation improves scalability, editability, and realism compared with global optimization systems.

Experiments on six room categories (living room, kitchen, bedroom, etc.) show quantitative and qualitative improvements over Holodeck and I-Design, evaluated through metrics such as CLIP similarity, object pose accuracy, semantic consistency, scene functionality, and aesthetics.

**Strengths:**

1. The paper presents a framework for text-driven 3D indoor scene generation. Its primary innovation lies in the agentic, stepwise generation paradigm, which allows the model to construct and edit 3D scenes incrementally rather than through one-shot global optimization. This paradigm closely resembles human iterative design workflows and enables flexible scene manipulation. The integration of multimodal large language models (MLLMs) with a rule-based constraint solver through the Reciprocal Rank Fusion (RRF) mechanism balances semantic constraints with visual spatial cues.

2. The paper demonstrates clear improvements over baselines (Holodeck, I-Design) across multiple room categories and metrics, supported by quantitative and qualitative results. The modular system design, which comprises room planning, visual layout reasoning, and numerical layout solving, is well explained, and the ablation studies provides valuable insights into the contribution of visual grounding.

**Weaknesses:**

1. Despite its well-structured design, the paper still has several weaknesses that limit its empirical depth and technical completeness. First, the evaluation methodology is overly dependent on MLLM-based scoring, without the inclusion of physical plausibility checks or human subjective assessments. As a result, it remains unclear whether the reported improvements genuinely reflect more realistic or functional layouts, or simply align with the evaluation model’s learned preferences. Incorporating physics-based constraints (e.g., collision detection, reachability analysis) or user studies assessing realism and aesthetics would substantially strengthen the experimental validation and credibility of the results.
2. Furthermore, the comparative analysis is relatively narrow. The experiments only benchmark against Holodeck and I-Design, omitting recent vision-language-based layout methods such as LayoutVLM (Sun et al., CVPR 2025), which also integrate visual cues and spatial relation reasoning. Including such baselines would offer a more comprehensive view of the field and provide stronger evidence of RoomGen’s advantages. In addition, the ablation study fixes the fusion weight α ≤ 0.5 and does not explore settings where visual cues are given higher priority. It remains uncertain whether larger α values could further improve performance or destabilize the optimization balance, and this sensitivity analysis is necessary to support the design choice.
3. Another major concern lies in the low spatial alignment between MLLM predictions and final object placements, which is only about 10% match closely. This discrepancy raises questions about how much the visual grounding actually contributes. A deeper analysis or visualization of these inconsistencies would help clarify the real role of the MLLM in layout generation.
4. The paper also lacks comprehensive qualitative evidence for several of its key claims. While incremental object insertion is demonstrated, most examples show additions near the boundary, without showing how the system behaves under different spatial constraint types (“in front of,” “side of,” “near,” etc.). Similarly, although the abstract emphasizes the system’s capacity for local rearrangement, no corresponding experiments or downstream applications are provided to verify this ability. The appendix includes only living room and bedroom examples, leaving other room types unexplored. Providing sequential visualizations for all categories would make the results more convincing.
5. Finally, the stepwise generation strategy, though flexible, introduces its own limitations. The authors acknowledge that early placement decisions can lead to “lock-ups” in dense scenes, yet the method lacks a backtracking or corrective mechanism to mitigate these errors. Incorporating an error recovery or re-planning step would significantly enhance robustness. Moreover, while the framework is described as “agentic,” its behavior is still largely rule-driven rather than autonomously learned. Introducing reinforcement learning or model-based optimization could enable genuine decision-making adaptivity and improve generalization to unseen environments. The claimed scalability of linear token growth with object count is also unsubstantiated by quantitative curves or statistics, and additional experiments showing the actual number of objects generated per scene and the token usage scaling trend would make this claim more credible.

**Questions:**

1. Many of the core evaluation metrics in this paper rely heavily on MLLM-based scoring, but there is a lack of physical metrics and human subjective evaluations to provide a more comprehensive assessment.

2. In the quantitative experiments, the paper should include comparisons with more text-to-scene generation methods that also integrate visual coordinate annotations or spatial relation reasoning, such as LayoutVLM: Differentiable Optimization of 3D Layout via Vision-Language Models [Sun et al., CVPR 2025].

3. Has the experiment been conducted with α values greater than 0.5? Would performance improve further under higher weighting of visual location cues?

4. Please explain why the low percentage of final layouts aligning with MLLM-predicted positions happens and how to improve this aspect.

5. Please include more qualitative experiments on incremental object insertion, not only along the room boundaries but also under the five types of spatial constraints described in Section 3.2.

6. In the appendix, please provide step-by-step qualitative results for all room categories beyond the living room and bedroom, showing sequential object placements.

7. The abstract mentions local rearrangement, but no related experiments or downstream tasks are shown. Please include qualitative results to demonstrate this capability.

8. The paper acknowledges that the stepwise approach can fail when the scene becomes highly occupied. It is suggested to incorporate an error detection or backtracking mechanism to mitigate such early-decision failures.

9. The paper claims that “our method can, in principle, populate scenes with arbitrarily many objects, and LLM token usage grows only linearly with the number of objects.” Please add experiments showing the average number of objects generated per scene under different settings, and provide a curve of token consumption vs. object count to substantiate this claim.

10. The paper lacks innovation in agentic decision-making mechanisms. It should consider introducing more autonomous learning components, such as reinforcement learning or model-based optimization, to enhance the agent’s adaptivity and autonomy.

---

### Official Review · Reviewer_6yYk · 2025-11-01

**Soundness:** 2
**Presentation:** 2
**Contribution:** 2
**Rating:** 2
**Confidence:** 5

**Summary:**

This paper proposes an agent-style, object-by-object framework for scene generation and editing. It first uses an LLM to produce the room type, dimensions, and an object list. At each step, a top-down view is rendered and an MLLM predicts both (i) semantic relational constraints and (ii) a suggested coordinate for the next object. A layout solver then converts the constraints into candidate poses and fuses them with the suggested coordinate via  RRF to select the final placement. The process repeats until all objects are placed. Experiments are conducted on multiple scenarios and show decent results. However, the method largely extends prior work; while the introduction of RRF is of some interest, the overall contribution is limited. The manuscript is also roughly written.

**Strengths:**

1. Sequential decisions conditioned on a top-down view naturally support insertion and local edits, avoiding full global recomputation and making the approach practical.
2. Effective fusion of semantics and geometry: the layout solver takes structured constraints together with a visual coordinate suggestion, and RRF provides a simple, effective, and reproducible trade-off that is easy to extend.
3. Using a gridded coordinate scheme is sensible and helps avoid the common issue that LLMs struggle to output precise continuous coordinates.

**Weaknesses:**

1. Performance is bounded by the underlying MLLM (train-free): as acknowledged by the authors, the ceiling depends on the base model. There is no end-to-end learnable mechanism for error correction or adaptation, and the pipeline lacks a systematic self-supervised remedy for misplacements or hallucinations.
2. The writing quality is rough, and several errors appear in figures and tables. For example, in Figure 1 the word “layout” is misspelled as “layoug.”
3. Although such methods do not necessarily require scaling up 3D assets, relying solely on HSSD feels insufficient. Assets from datasets like 3D-FRONT and those used in Holodeck should also be tested.
4. The methodological novelty is limited and not fundamentally different from prior work. For indoor scene generation, several important aspects are missing: placement of small objects, handling of irregular floor shapes, and incorporation of richer, finer-grained rules.

**Questions:**

1. How long does it take to generate a room, and how many tokens are consumed? End-to-end generation time and token count might be acceptable for a single pass, but the step-by-step setting needs more detailed reporting.
2. How strong is instruction following? If a user specifies spatial relations directly in the input, what results do you obtain?
3. In principle, a step-by-step approach should perform better in irregular rooms. Can you evaluate this case?

---

### Meta-Review · Area_Chair_EUDS · 2026-01-05

**Summary:**

The paper presents an interesting application of MLLMs for scene generation, but the limited novelty, problematic evaluation protocols, missing baselines, and unresolved technical limitations regarding error recovery make it unsuitable for publication at ICLR. The absence of an author response prevented any clarification or defense of these issues.

**Reviewer Concerns:**

Since there was no rebuttal, all concerns remain valid. The rejection is based on the following consensus points:

Limited Novelty: The method is largely an extension of prior work (like Holodeck) using off-the-shelf LLMs rather than a fundamentally new approach.

Flawed Evaluation: The metrics rely too heavily on MLLM/CLIP scoring (which is non-standard and potentially unreliable for this task) rather than human evaluation or physics-based checks.

Missing Baselines: Failure to compare against relevant state-of-the-art methods like LayoutVLM and Direct Numerical Layout Generation.

Other technical issues such as ineffective visual grounding.

**Reviewer Scores:**

Unlikely to get higher scores due to lack of rebuttal and all the reviewers unanimously voted for rejection.

---

### Decision · Program_Chairs · 2026-01-26

Reject